# A population-based cohort study of socio-demographic risk factors for COVID-19 deaths in Sweden

Sven Drefahl [1✉], Matthew Wallace [1], Eleonora Mussino [1], Siddartha Aradhya [1], Martin Kolk [1,2], Maria Brandén [1,3], Bo Malmberg [4] & Gunnar Andersson [1]

As global deaths from COVID-19 continue to rise, the world's governments, institutions, and agencies are still working toward an understanding of who is most at risk of death. In this study, data on all recorded COVID-19 deaths in Sweden up to May 7, 2020 are linked to high-quality and accurate individual-level background data from administrative registers of the total population. By means of individual-level survival analysis we demonstrate that being male, having less individual income, lower education, not being married all independently predict a higher risk of death from COVID-19 and from all other causes of death. Being an immigrant from a low- or middle-income country predicts higher risk of death from COVID-19 but not for all other causes of death. The main message of this work is that the interaction of the virus causing COVID-19 and its social environment exerts an unequal burden on the most disadvantaged members of society.

[1] Stockholm University Demography Unit (SUDA), Department of Sociology, Stockholm University, Stockholm, Sweden. [2] Institute for Futures Studies, Stockholm, Sweden. [3] Institute for Analytical Sociology (IAS), Department of Management and Engineering, Linköping University, Norrköping, Sweden. [4] Department of Human Geography, Stockholm University, Stockholm, Sweden. ✉email: sven.drefahl@sociology.su.se

As global deaths from COVID-19 continue to rise[1], the world's governments, institutions, and agencies are still working toward an understanding of who is most at risk of death. This is due to a lack of high-quality microlevel data linking death records to other data sources (e.g., censuses, surveys, and registers) that contain information on sociodemographic background characteristics associated with variation in the risk of death. Until now, our understanding has been limited to rudimentary inferences drawn from comparisons of counts or proportions for different sections of society, more formal analyses of highly aggregated data, and a small number of microlevel analyses focused on comorbidities that give scant attention to sociodemographic factors beyond age and sex. These initial findings suggest that men[2–6], the elderly[2–11], racial and ethnic minorities[4,6,7,12], and people occupying lower socioeconomic positions[4,6], are more prone to developing severe COVID-19, or dying from it. In light of the widely adopted message that COVID-19 "does not discriminate", one given credence by the Director-General of the World Health Organization[13], these patterns have been met with alarm. Indeed, they have spurred calls for the release of more detailed case and fatality data that would permit a more rigorous investigation of the apparent sociodemographic inequalities in COVID-19 death[8]. In response to this call, the present study examines sociodemographic risk factors of COVID-19 mortality in Sweden.

Owing to the special provision of new data from the Swedish authorities, we have access to data on all recorded COVID-19 deaths in Sweden up to May 7, 2020 linked to high-quality and accurate individual-level background data from administrative registers. Using these data, we aim to advance the understanding of the sociodemographic risk factors associated with the risk of COVID-19 death for the entire population of Sweden.

As many nations now have begun to ease distance restrictions and plan a roadmap through the pandemic, the provision of reliable information on which members of society are most at risk of death will be essential to informing national strategy. Compared to most other nations, Sweden has taken a less-restrictive approach to containing COVID-19 by encouraging working from home and promoting social distancing rather than mandating quarantine, while relying on a high level of compliance of its population to these measures from the very beginning. Kindergarten and schools have remained open for children below age 16 throughout the pandemic, while upper secondary schools and universities were closed on March 17 (ref. [14]). While Sweden has experienced relatively high levels of COVID-19 mortality per capita[1], its experience may provide invaluable insight for other countries to prepare for upcoming developments. As such, the results from the present study are not only important for the Swedish context, but also informative for other contexts to identify vulnerable populations and the circumstances in which they are at higher risk.

Here, we examine how the risk of death varies across fundamental sociodemographic characteristics, including age, sex, civil status, individual disposable income, region of residence, and country of birth. Our results reveal an additional burden on the most vulnerable individuals of society that should be of interest to decision makers in all countries.

## Results

During the 1,189,484 person-years of observation, 17,181 deaths occurred in our study population between March 13, 2020 and May 7, 2020. Table 1 shows the distribution of population at risk, and deaths from COVID-19 and all other causes of deaths for all variables used in our analyses, as well as categorized in broad age groups. A more detailed stratification of deaths and time at risk by age group and sex is presented in Supplementary Tables 1 and 2. Table 1 shows an increasing risk of dying from COVID-19 by increasing age, as well as excess mortality for men, widowed, primary educated, those with low income, those from high-income countries (HIC), and those living in Stockholm county. For all other causes of death similar patterns were mostly found, except for the county of residence; those living outside Stockholm experience higher crude mortality rates than those in Stockholm.

Figure 1 compares the mortality risks from COVID-19, separately for men and women, controlled for age from a multivariate Cox survival analysis. All point estimates and confidence intervals from these models can be found in the corresponding Supplementary Table 3. Figure 1 shows, for men and women alike, that never married, divorced, and widowed individuals experience ~1.5–2 times higher mortality from COVID-19 than those who are married. We consider socioeconomic position as measured by both education and individual net income for both sexes. With respect to education, and net of income, we find a gradient for both men and women with individuals with secondary education experiencing 25% ($HR_{Men}$: 1.25; 95% CI: 1.09, 1.43) and 38% ($HR_{Women}$: 1.38; 95% CI: 1.17, 1.62) higher mortality, respectively, than individuals with postsecondary education, and those with primary education experiencing 24% ($HR_{Men}$: 1.24; 95% CI: 1.07, 1.43) and 51% ($HR_{Women}$: 1.51; 95% CI: 1.28, 1.79) higher mortality, respectively, relative to the same reference group. We also find a pronounced income gradient, net of education, for men but not for women. Among men, individuals in the first and second tertiles of individual net income experiencing ~75% ($HR_{Q1}$: 1.76; 95% CI: 1.49, 2.09) and 50% ($HR_{Q2}$: 1.51; 95% CI: 1.29, 1.78) higher mortality, respectively, relative to those in the top tertile. Among women, those in the first tertile experiencing 26% ($HR_{Q1}$: 1.26; 95% CI: 1.01, 1.58) higher mortality, while those in the second tertile experience the same mortality ($HR_{Q2}$: 0.99; 95% CI: 0.78, 1.25) compared to those in the top tertile. Immigrants from low- and middle-income countries from the Middle East and Northern Africa displayed more than three times higher mortality among men (HR: 3.13; 95% CI: 2.51, 3.90) and two times higher among women (HR: 2.09; 95% CI: 1.52, 2.89), as compared to those born in Sweden, ceteris paribus; whereas immigrants from other low- and middle-income countries experienced more than twice as high mortality among men (HR: 2.20; 95% CI: 1.81, 2.69) and ~1.5 times higher mortality among women (HR: 1.45; 95% CI: 1.12, 1.90). Immigrants from HIC only displayed 19% higher mortality among men (HR: 1.19; 95% CI: 1.01, 1.39) and 8% higher mortality among women (HR: 1.08; 95% CI: 0.92, 1.26). Finally, the variable that has the strongest impact on COVID-19 mortality is living in Stockholm County, as compared to living in the rest of Sweden, which is associated with ~4.5 times higher mortality.

Several of the gradients in COVID-19 mortality are similar to those observed for mortality from all other causes of death (Supplementary Table 3). This holds for variables related to sociodemographic characteristics, but not for those related to geographical factors. For men, living in Stockholm was associated with 4.5 times higher risk of dying from COVID-19 (HR: 4.51; 95% CI: 4.08, 5.00), but only a 1.2 times higher risk of dying from all other causes of death (HR: 1.24; 95% CI: 1.16, 1.31). The differences by country of origin are even more compelling. Mortality from all causes of death reflect the healthy migrant advantage, with male migrants from low- and middle-income countries in Northern Africa and the Middle East exhibiting 18% lower mortality than native Swedes (HR: 0.82; 95% CI: 0.68, 0.99). For women, we find similar differences in gradients between COVID-19 mortality and other causes of death, by county of residence and country of birth.

**Table 1 Observations, deaths, and exposure time to the risk of death from COVID-19, and all other causes of death in Sweden (March 13, 2020–May 7, 2020).**

| | Observations on March 12 | | Exposure time | Deaths | | | | | |
| --- | --- | --- | --- | --- | --- | --- | --- | --- | --- |
| | | | | COVID-19 | | | All other causes | | |
| | N | % | Years | N | % | Rate per 1000 | N | % | Rate per 1000 |
| Age | | | | | | | | | |
| 20–49 | 3,830,151 | 49.3 | 587,542 | 30 | 1.0 | 0.05 | 358 | 2.5 | 0.61 |
| 50–69 | 2,393,619 | 30.8 | 366,583 | 264 | 8.4 | 0.72 | 1883 | 13.4 | 5.14 |
| 70–79 | 999,369 | 12.9 | 152,406 | 689 | 22.0 | 4.52 | 3228 | 23.0 | 21.18 |
| 80–89 | 447,371 | 5.8 | 67,553 | 1328 | 42.5 | 19.66 | 5041 | 35.9 | 74.62 |
| 90+ | 104,544 | 1.3 | 15,401 | 815 | 26.1 | 52.92 | 3545 | 25.2 | 230.18 |
| Sex | | | | | | | | | |
| Men | 3,876,881 | 49.9 | 593,068 | 1683 | 53.8 | 2.84 | 7014 | 49.9 | 11.83 |
| Women | 3,898,173 | 50.1 | 596,416 | 1443 | 46.2 | 2.42 | 7041 | 50.1 | 11.81 |
| Civil status | | | | | | | | | |
| Married | 3,302,934 | 42.5 | 505,974 | 1032 | 33.0 | 2.04 | 4654 | 33.1 | 9.20 |
| Never married | 3,059,130 | 39.3 | 467,565 | 348 | 11.1 | 0.74 | 1947 | 13.9 | 4.16 |
| Divorced | 993,609 | 12.8 | 152,106 | 598 | 19.1 | 3.93 | 2479 | 17.6 | 16.30 |
| Widowed | 419,381 | 5.4 | 63,839 | 1148 | 36.7 | 17.98 | 4975 | 35.4 | 77.93 |
| Education | | | | | | | | | |
| Primary | 1,307,280 | 16.8 | 198,679 | 1224 | 39.2 | 6.16 | 6096 | 43.4 | 30.68 |
| Secondary | 3,439,340 | 44.2 | 526,819 | 1183 | 37.8 | 2.25 | 5265 | 37.5 | 9.99 |
| Postsecondary | 2,891,664 | 37.2 | 443,122 | 577 | 18.5 | 1.30 | 2416 | 17.2 | 5.45 |
| Missing | 136,770 | 1.8 | 20,864 | 142 | 4.5 | 6.81 | 278 | 2.0 | 13.32 |
| Individual net income | | | | | | | | | |
| Tertile 1 (low) | 2,592,086 | 33.3 | 395,307 | 1916 | 61.3 | 4.85 | 9024 | 64.2 | 22.83 |
| Tertile 2 | 2,593,056 | 33.4 | 397,225 | 894 | 28.6 | 2.25 | 3598 | 25.6 | 9.06 |
| Tertile 3 (high) | 2,589,912 | 33.3 | 396,952 | 316 | 10.1 | 0.80 | 1433 | 10.2 | 3.61 |
| Country of birth | | | | | | | | | |
| Sweden | 6,184,398 | 79.5 | 946,008 | 2406 | 77.0 | 2.54 | 12,193 | 86.8 | 12.89 |
| HIC | 521,804 | 6.7 | 79,844 | 363 | 11.6 | 4.55 | 1212 | 8.6 | 15.18 |
| LMIC other | 680,369 | 8.8 | 104,151 | 195 | 6.2 | 1.87 | 444 | 3.2 | 4.26 |
| LMIC MENA | 388,483 | 5.0 | 59,480 | 162 | 5.2 | 2.72 | 206 | 1.5 | 3.46 |
| County of residence | | | | | | | | | |
| Other | 6,017,966 | 77.4 | 920,692 | 1569 | 50.2 | 1.70 | 11,337 | 80.7 | 12.31 |
| Stockholm | 1,757,088 | 22.6 | 268,792 | 1557 | 49.8 | 5.79 | 2718 | 19.3 | 10.11 |
| Total[a] | 7,775,054 | 100 | 1,189,484 | 3126 | 100 | | 14,055 | 100 | |

*HIC* high-income countries, *LMIC MENA* low-middle-income countries from Northern Africa and the Middle East, *LMIC other* other low-middle-income countries.
[a]Sum of exposure time over all categories may not always add up to the total because of rounding.

Figure 2 presents the results comparing mortality risks from COVID-19 stratified for individuals in working age (ages 65 and below) and retirement age (ages 66 and higher) in a multivariate setup that also includes continuous age within each age segment. Due to the lower number of deaths in the younger age group, the confidence intervals for the estimated death risks of this strata are much larger than for the elderly. In this analysis, we still find substantial differences in the demographic risk factors between working age people and retirees. Among individuals in working ages, as compared to retirees, males experience even higher mortality relative to females, and education and income are stronger predictors of dying from COVID-19. In working ages, those in the lowest income tertile are more than five times as likely to die (HR: 5.40; 95% CI: 3.51, 8.35) from COVID-19 than those in the highest tertile, and those with primary (HR: 2.62; 95% CI: 1.65, 4.16) and secondary (HR: 2.22; 95% CI: 1.46, 3.37) education are more than twice as likely to die relative to those with postsecondary education. Among individuals in retirement ages, the socioeconomic differentials in COVID-19 mortality are much less pronounced, this holds both for the independent roles of educational attainment and (pension) income. In contrast to those in working ages, we find more pronounced differences by civil status among the elderly. Specifically, the death risks relative to married individuals were 48% higher among widows and

widowers (HR: 1.48; 95% CI: 1.34, 1.63), 62% higher for individuals who were divorced (HR: 1.62; 95% CI: 1.46, 1.80), and 65% higher for those never married (HR: 1.65; 95% CI: 1.44, 1.89). In the working age population, significant excess mortality in COVID-19 is only observed for those who were never married (HR: 1.48; 95% CI: 1.04, 2.10). Immigrants from low and middle-income countries are approximately twice as likely to die, as compared to individuals born in Sweden in both age segments. The excess mortality associated with living in Stockholm is also very similar in the two age groups.

Again, the contrast to mortality patterns for all other causes of death is instructive (see also Supplementary Table 4). We observe once more a differentiation between the roles of socio-demographic and geographical factors in COVID-19 and all-cause mortality. In both age segments, the gradients of most sociodemographic variables are largely similar for all other causes of death and COVID-19 mortality. Again, the patterns for country of birth and county of residence are entirely different for all-cause and COVID-19 mortality. For further reference, results from univariate Cox regressions (age adjusted) and results from a fully adjusted model for the combined study population are presented in Supplementary Table 5, for COVID-19 deaths, and Supplementary Table 6, for all other causes of death. In order to ensure lack of multicollinearity, we also examined the correlation

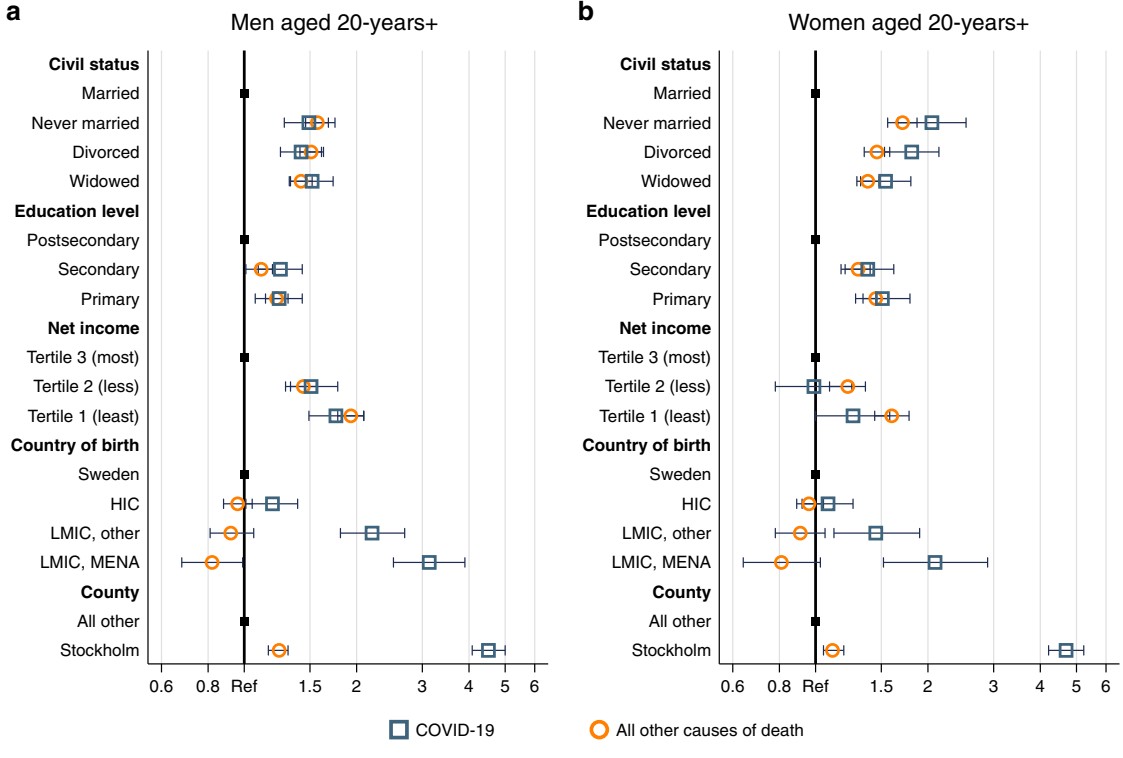

**Fig. 1 Hazard ratios of dying from COVID-19 and all other causes of death for men and women in Sweden. a** Men aged 20 years and older, error bars representing 95% confidence intervals of hazard ratios, $n = 3,876,881$ men. **b** Women aged 20 years and older, error bars representing 95% confidence intervals of hazard ratios, $n = 3,898,173$ women. Blue squares indicating COVID-19 mortality. Orange circles indicating mortality from all other causes of death.

matrix of the coefficients of the Cox model. The highest correlated coefficients ($R = 0.3$) we found for being in the lowest income tertile and having primary education, suggesting that multicollinearity is not a problem in our analyses. The proportional hazards assumption was also tested and sex, individual income, and education were found to violate the assumption in some cases, which could potentially lead to biased parameter estimates. The age and sex-stratified models presented earlier address some of these violations. Further robustness checks in the form of stratified analyses by sex, individual income, and education did not significantly influence other parameter estimates.

## Discussion
We provide a comprehensive study of sociodemographic risk factors of COVID-19 death, using complete, detailed, and high-quality microlevel data for an entire national population. Specifically, in addition to being in older ages, we show that being male, having low (or no) individual income, having a low education level, not being married, and being born in a low- or middle-income country all independently predict a higher risk of death from COVID-19. We also find important differences in the patterns between the working age and retirement age populations, in the more pronounced role of socioeconomic characteristics at working ages and the more pronounced role of civil status at retirement ages. Our findings have direct relevance to Sweden and important implications for other Western countries that are easing lockdown restrictions in favor of the more open strategy adopted by Sweden from the outset. Our study provides valuable insight for countries preparing for the challenges associated with trying to live with the virus, and a better understanding of

potential alternative scenarios for countries, with similar health care and welfare systems.

Previous research shows that people with less strong socio-economic resources are more vulnerable to poor health and mortality from different causes of death, as we find here[15–17]. However, more research is needed to understand all the mechanisms behind low-income and low-educated individuals' excess COVID-19 mortality. Preexisting health conditions are likely to matter; precarious employment and crowded housing are other factors that could help understand this relationship. Future research on COVID-19 mortality also needs to pay better attention to foreign-born people. Research is needed to understand to which extent their elevated COVID-19 mortality is related to factors, such as transnational activities, living arrangements, occupations, and the characteristics of the neighborhoods in which they live. Of particular interest would be to understand if migrants are more susceptible to severe outcomes of the disease once infected, or if excess mortality is related to transmission pathways. Here, we demonstrate that we find a migrant disadvantage after adjusting for socioeconomic characteristics and region of residence in Sweden.

For people in pension age, the Swedish policy has been directed to reduce intergenerational contact. The number of multi-generational households in Sweden is relatively low[18]. Our results show that unmarried elderly people are at particularly high risk of dying from COVID-19. This is the segment of the population that is in higher need than others to rely on external assistance in their home, or who lives in a care home. Future research needs to study in more detail how different characteristics of old people's housing arrangements, including elderly care and home-help services, have been related to COVID-19 mortality.

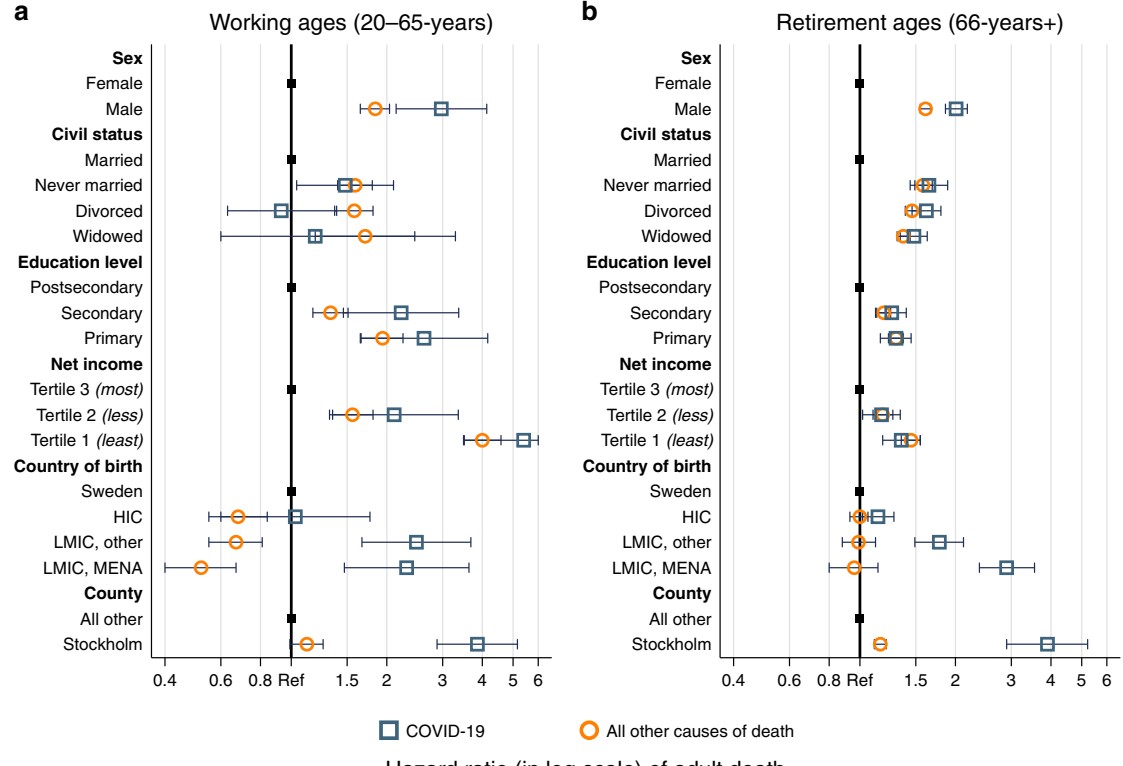

**Fig. 2 Hazard ratios of dying from COVID-19 and all other causes of death for working ages and retirement ages. a** Working ages (20–65 years), error bars representing 95% confidence intervals of hazard ratios, n = 5,813,359 individuals. **b** Retirement ages (66 years and older), error bars representing 95% confidence intervals of hazard ratios, n = 1,979,710 individuals. Blue squares indicating COVID-19 mortality. Orange circles indicating mortality from all other causes of death.

A major strength of our study is that we have complete coverage of the total population and all deaths in Sweden, from both COVID-19 and other causes. Thus, our analysis does not suffer from selection into our study population. However, mortality from other causes, that we use as comparison, is occurring in a setting where COVID-19 exists, and thus may capture indirect COVID-19 effects (collateral deaths). For instance, the fear of contracting COVID-19 may impact the care seeking of individuals, which may in turn increase the risk of other causes of death. As such, by only studying confirmed COVID-19 deaths, we may underestimate the true mortality effect of the pandemic. Also, although the Swedish data on COVID-19 deaths is considered accurate, we cannot rule out some misclassification of COVID-19 deaths.

Our relatively short follow-up period of slightly <2 months is a potential limitation, as we cannot take full advantage of the time to event information, one of the primary strengths of Cox regression. Potentially, Cox regression may even produce bias if there is substantive unaccounted spatiotemporal variation in the spread of the disease. To assess this possibility, we estimated logistic regression models by gender (Supplementary Table 7), which ignore the timing of the event and thus may rule out the possibility that the findings are driven by such patterns. We found no substantive difference between the Cox and the logistic regression. A second limitation is that we do not have information on emigration from Sweden during the first 5 months of 2020. Consequently, we may underestimate mortality rates to a small extent. However, the impact on our results is likely marginal as only 7% of all emigrations in recent years occur at ages with substantive risks of dying from COVID-19 (60+).

Importantly, we show that the interaction of the virus and its social environment exerts an unequal burden on the most disadvantaged members of society. Beyond the strong effects of age on COVID-19 mortality, we find that disadvantaged sub-populations show elevated mortality risk—as is the case for most other causes of death and mortality in general[19,20], particularly so for individuals in working age. However, the robust finding of elevated mortality among immigrants from low and middle-income countries deviates from findings on other causes of death, such as neoplasms or circulatory diseases, where immigrants tend to have lower mortality than natives[21]. This finding may help guide governmental policy, and may be of relevance for other HIC as well. Better health care resources may need to be allocated toward communities with many foreign-born individuals.

## Methods

**Data and study population.** The data for this study are individual-level Swedish register data collected and maintained at various state agencies, and combined and stored for research purposes at Statistic Sweden's secure data storage facility. The data were accessed through the micro-online access system MONA[22]. The data for this study are 7,943,843 individuals aged 20 and above on March 12, 2020, and who were living in Sweden in December 2019. This age restriction was established because the youngest individual in our data to die from COVID-19 was 20 years old (Supplementary Fig. 3). The follow-up period was March 13 up until May 7, 2020. The flow diagram showing the inclusion and exclusion criteria is presented in Supplementary Fig. 1. We excluded individuals who had not lived in Sweden in the two prior years (N = 147,557) because those could not be linked to all records of data. We also excluded a small percentage of individuals for whom we had no registered information on country of birth (N = 8370) and income (N = 12,862). The final study population consists of 7,775,064 individuals with an average follow-up time of 56 days amounting to a total of 1,189,484 person-years under observation.

**Outcome.** Our outcome variable is defined from all deaths reported between March 13, 2020 (the date of the first confirmed death from COVID-19 in our data) and May 7, 2020, and whether each death was associated with COVID-19. All deaths have been collected by the Swedish National Board of Health and Welfare,

the agency responsible for the cause of death register, and match the death counts reported to the public. The distribution of COVID-19 deaths for each day of our observation period are presented in Supplementary Fig. 2. In the study population, a total of 17,181 individuals died during the studied period; 3126 of these deaths were reported to be from COVID-19. In 2988 of these cases, COVID-19 was identified as the underlying cause of death (emergency ICD code U07.1, U07.2, or B34.2). In the remaining 138 cases, ICD emergency codes U07.1, U07.2, or B34.2 were listed as contributing causes of death, but not as the underlying cause of death. Given the timeliness of the data, the assignment of the underlying cause of death should be understood as preliminary.

**Independent variables**. Thanks to the unique personal identification number of every person with legal residence in Sweden, each individual COVID-19 death was linked to data from a collection of Swedish administrative registers, which includes individual sociodemographic information for all individuals ever living in Sweden during 1968–2019. From the total population register, we derived information on country of birth, sex, age, place of residence, and civil status. Country of birth is classified according to the World Bank classification based on the Gross National Incomes per capita using the World Bank Atlas method[23], as (i) Sweden, (ii) HIC, (iii) low-middle-income countries from Northern Africa and the Middle East (LMIC MENA), and (iv) other low-middle-income countries (LMIC other). Place of residence is categorized into Stockholm county versus the rest of Sweden. Civil status is classified into (1) never married, (2) married, (3) widowed, and (4) divorced. We derived data on individual net income and educational attainment from the Longitudinal integrated database for health insurance and labor market studies (LISA)[24]. Individual net income is categorized into tertiles based on all adult residents of Sweden, while education is categorized into: (i) primary schooling (9 or fewer years of education), (ii) secondary education (10–12 years of education), (iii) postsecondary education (>12 years of education), and (iv) those with missing information on education. The vast majority of those with missing information on education (~88%) are immigrants or special categories of elderly, who did not have any formal education or failed to report their education to the Swedish authorities. As such, this category is potentially associated with the risk of dying from COVID-19 and was therefore not removed from the analysis.

**Statistical analysis**. Cox proportional hazard regression models were estimated to obtain hazard ratios for the risk of dying between March 13, 2020 and May 7, 2020. The underlying time process of the Cox model is biological age measured with monthly precision. Early research in the pandemic identified age and sex as two of the most important predictors for the risk of dying. It is therefore plausible that age and sex moderates the impact from socioeconomic variables on mortality risks. In order to account for this, we estimated two sets of models. The first set presents results for women and men, separately, the second set presents separate results for individuals in working age (ages 65 and below) and retirement age (ages 66 and higher). In each set, we fitted separate regressions estimating (1) the cause-specific hazard of dying from COVID-19, right-censoring individuals who die from other causes, and (2) the cause-specific hazard of dying from other causes than COVID-19, right censoring at death from COVID-19. The correlation matrices of the parameter estimates were obtained for each model. The proportional hazards assumption was tested on the basis of Schoenfeld residuals and log–log plots. Alternative methods for handling ties were tested, and in sensitivity analyses logistic regressions were estimated (Supplementary Table 7). All of the covariates are time constant and either measured at the end of 2019 (sex, marital status, country of birth, and living in Stockholm) or 2018 (highest achieved educational degree and individual net income). We adjust for whether an individual lived in Stockholm, since it was the epicenter of the virus in Sweden. All analyses were conducted using Stata Statistical Software: Release 16 (StataCorp LP, College Station, Texas).

**Reporting summary**. Further information on research design is available in the Nature Research Reporting Summary linked to this article.

## Data availability
This study is produced under the Swedish Statistics Act, where privacy concerns restrict the availability of register data for research. Aggregated data can be made available by the authors, conditional on ethical vetting. The authors access the individual-level data through Statistics Sweden's micro-online access system MONA. The authors linked data from the "Historical Population Register (HBR)", the "Register of the Total Population (RTB)", the "Cause of Death Register", and the "Longitudinal integrated database for health insurance and LISA". More information about data availability and data access can be obtained from: https://www.scb.se/en/services/guidance-for-researchers-and-universities/mona–a-system-for-delivering-microdata/. The analyses have been approved by the Swedish ethical-vetting authority, Dnr 2020-02199.

## Code availability
Computer code is available from Figshare under https://su.figshare.com/articles/Socio-demographic_risk_factors_of_COVID-19_deaths_in_Sweden_A_nationwide_register_study/12420347.

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

## Acknowledgements

We are grateful for financial support from the Swedish Research Council for Health, Working Life, and Welfare (FORTE) for a project with ageing research, grant number 2016-07115. The funders of the study had no role in the study design, data collection, data analysis, data interpretation, or writing of the report. We thank Thomas Niedomysl of Region Halland, Petra Westin of the National Board of Health and Welfare, and Simon Kurt of Statistics Sweden for their invaluable role in providing the data. We thank Emma Pettersson and Jesper Lindmarker for their help with editing the manuscript. We also thank our babies and toddlers—Cecilia Wallace, Kanai Aradhya, Charlie Söderlund Brandén, Ella, and Vera Mussino Drefahl—for their patience with their parents in home office during the COVID-19 pandemic. The corresponding author had full access to all the data in the study and had final responsibility for the decision to submit for publication.

## Author contributions

S.D. jointly conceived the study with E.M., S.A., and M.W. G.A., B.M., and M.B. provided the data. S.D. analyzed the data. S.D., M.W., E.M., S.A., and M.K. wrote the manuscript; G.A. and M.B edited the manuscript. S.D., with the help of E.M., took the lead in the revision process. G.A. supervised the project. All the authors read and approved the final version of the manuscript.

## Funding

## Competing interests

The authors declare no competing interests.
