## [Peer Review File · Nature Communications]

REVIEWER COMMENTS

Reviewer #1 (Remarks to the Author):

This paper used register-data for the entire population of Sweden in order to investigate sociodemographic risk factors of Covid-19 deaths. The study is novel and well conducted with results that strongly suggest that Covid-19 does discriminate with respect to sociodemographic factors beyond age and sex. I have the following major and minor comments and suggestions to the authors:

Major comments

1. Descriptive data

Sufficient descriptive data are lacking in order to assess the validity of the results and to get a deeper understanding of what is going on behind the presented multivariable analyses. In addition to Supplementary Table 1, I would suggest a case-control descriptive table with detailed stratification by sex and age, e.g. 18 – 49, 50 – 69, 70 – 79, 80 – 89 and 90+. Such a table should show background data (civil status, education, income, country of birth and county of residence) with separate columns for persons at risk and persons who died from Covid-19 during follow up. This presentation would elucidate for a given sex and age how the other factors are related to the risk of Covid-19 death.

2. Cox regression

The choice of Cox regression with biological age as underlying time scale can be questioned as the length of follow up is so short (all cases occur during 2 months) that the timing of the event (early vs. later deaths) is not really relevant to take into account (as opposed to studies where the follow up is years or even decades). The time-to-event analysis could also lead to bias due to initial spatio-temporal variation in the spread of the disease, for example if the disease hit elderly care homes with many foreign-born residents first and then elderly care homes with mostly Swedish-born residents a month later. I would therefore suggest that logistic regression (which ignores the timing of event) is used as a sensitivity analysis to rule out the possibility that the findings are driven by such patterns. Additionally, day 1-35 and 36 – 65 since the first Covid-19 death could be analysed separately to check for differential patterns in the reported relative risks that could be related to the spread of the disease.

3. Competing risks

The paper lacks a description of how deaths from other causes are handled. In sensitivity analyses, the authors could also consider assessing potential biasing effects of competing deaths.

4. Associations with deaths from other causes

Multivariable analyses of associations with deaths from other causes during the same follow up period, similar to the ones presented in Figure 2 and Supplementary Table 2, would strengthen the discussion of how specific the observed associations are with deaths in Covid-19.

Minor comments

1. Line 53: "The data for this study are 3 135 deaths..."

I would rather say (line 66-67) that the

The data for this study are 6.96 million individuals (minus those with missing information on key variables) aged 18 and above, of whom 3 122 died from Covid-19 during follow up.

2. Finally, I would check to what extent the associations are the same in "Other counties" (as group) as in Stockholm.

Jonas Björk, Lund University, Sweden

Reviewer #2 (Remarks to the Author):

This study used administrative population-wide databases in Sweden to examine social and demographic factors associated with the risk of dying from Covid-19 between March and May 2020. The topic is important providing data on health inequalities associated with the Covid-19 pandemic. I have some concerns mainly with the data analysis and presentation of results. I would recommend the authors consult the STROBE statement when revising the manuscript <https://www.strobe-statement.org/>

1. It would be helpful to include some quantitative results (HRs and confidence intervals) in the abstract for the main findings reported on lines 10-11.
2. More information should be provided at the start of the methods on the data sources including links. For example, that the data was accessed via Mona with a link or reference. This is provided in the reporting summary document but is not explicit in the manuscript.
3. Description of the eligible population and inclusion criteria (adults aged 18 years and resident in Sweden for 2 years) should be come sooner in the methods section with justification for the inclusion and exclusion decisions taken.
4. All variables included in the analysis need to be more clearly defined and the authors should give the sources and details of measurement of each of them. Which Swedish administrative registers were used for the data linkage and how were the sociodemographic factors measured and recorded in those registers?
5. The statistical analysis should also present descriptive results including the characteristics of the participants and information on the exposure variables and a summary of follow-up time.
6. The statistical method needs to included statements on a priori decisions taken for the cox proportional hazard models, such as the reasons why the models were stratified on sex and age. What was the evidence that sex was an effect modifier? Overall analyses should be presented, not just for the sub-groups.
7. There is no information provided on model fit or diagnostics. Sociodemographic variables are correlated and so multicollinearity needs to be checked.
8. Unadjusted as well as adjusted estimates should be reported.
9. The figures should include a column reporting the actual effect estimates and confidence intervals (with p-values for the overall association for each variable). Alternatively, the authors could consider replacing the figures with tables as it is difficult to read directly from the figures.
10. There is no reference made in the main manuscript to the supplementary figure. Some explanation of its relevance should be included – e.g. to support the choice of time period studied.
11. The paragraph starting at line 164 in the discussion considers other social determinants but there are no references to existing literature.
12. Limitations of the study are not addressed in the discussion.
13. I feel more could be discussed in terms of Sweden's experience with Covid-19 and the approach taken by Sweden to manage Covid-19 and how this underpins the findings of this study. Some description of Sweden's approach in more detail rather than just stating the "more open strategy adopted by Sweden from the outset". Readers will be interested in these results because of the Swedish approach. This is an international journal therefore some more context is needed.

Claudia Slimings

Dear Reviewers,

Thank you very much for providing crucial and very sensible comments and suggestions for improving the manuscript. Consequently, we made substantive changes to the “Materials and Methods” section, added more descriptive statistics both to the main body of the manuscript as well as the supplementary material, added new analyses for all other causes of death, and also conducted a number of sensitivity analyses, such as logistic regression. The paper now follows the items laid out by the STROBE statement, for example we indicate the study design already in the new manuscript title “Socio-demographic risk factors of COVID-19 deaths in Sweden: A population-based cohort study”.

To allow a meaningful comparison between COVID-19 mortality and all other causes of deaths, we changed the start of our observation period from Jan 1 to March 13, which is the day of the first observed death from COVID-19 in Sweden. We also received the correct date of death for the person who initially died first in our data on March 5 (5/3). The correct date of death is May 3 (3/5). Consequently, we updated all figures and took the opportunity to also change the entry age into the study population to age 20, which is the lowest adult age of death in our data. Lastly, we added a new group to the country of birth variable, which now distinguishes low-middle income countries into low-middle income countries from Northern Africa and the Middle East (MENA) and all other low-middle income countries. This is because of the intense focus in the Swedish public on the particular group of migrants from MENA countries. Overall, we feel that the manuscript has improved considerably.

In the following please find our replies to each point:

Reviewer #1

1. Descriptive data

Sufficient descriptive data are lacking in order to assess the validity of the results and to get a deeper understanding of what is going on behind the presented multivariable analyses. In addition to Supplementary Table 1, I would suggest a case-control descriptive table with detailed stratification by sex and age, e.g. 18 – 49, 50 – 69, 70 – 79, 80 – 89 and 90+. Such a table should show background data (civil status, education, income, country of birth and county of residence) with separate columns for persons at risk and persons who died from Covid-19 during follow up. This presentation would elucidate for a given sex and age how the other factors are related to the risk of Covid-19 death.

Following the advice of both reviewers we have extended the descriptive tables. First, we extended the main descriptive table with mortality from all other causes and moved it from the appendix into the main text where we also describe its content. Second, we constructed descriptive tables by age and sex for each covariate as suggested by the reviewer. One table for men (Table S1) and one for women (Table S2) can now be found in the supplementary material. Both are also referred to in the text.

2. Cox regression: The choice of Cox regression with biological age as underlying time scale can be questioned as the length of follow up is so short (all cases occur during 2 months) that the timing of the event (early vs. later deaths) is not really relevant to take into account (as opposed to studies where the follow up is years or even decades). The time-to-event analysis could also lead to bias due to initial spatio-temporal variation in the spread of the disease, for example if the disease hit elderly care homes with many foreign-born residents first and then elderly care homes with mostly Swedish-born residents a month later. I would therefore suggest that logistic regression (which ignores the timing of event) is used as a sensitivity

analysis to rule out the possibility that the findings are driven by such patterns. Additionally, day 1-35 and 36 – 65 since the first Covid-19 death could be analysed separately to check for differential patterns in the reported relative risks that could be related to the spread of the disease.

We agree and understand that a number of different regression approaches are applicable for our data, including logistic regression and poisson regression. Following your advice we conducted logistic regression analysis as sensitivity check and added the results as a supplementary table (Table S7). Parameter estimates between Cox regression and logistic regression should be almost directly comparable as the risk to die in the study population is generally still very low and thus Odds-Ratios based on probabilities are very similar to Relative-Risks based on (incidence) rates of death. We conclude that there is hardly any difference between the results of the two models. Following your comment, we added a brief discussion of this issue in the limitation section. However, we still think that hazard regression is the more appropriate approach for our data with time information measured with daily precision, and thus we decided that it will remain our main analytical approach. We additionally conducted analyses separating early deaths vs not early deaths to test for differential patterns. The results showed that mortality differentials were slightly smaller in the second half of the observation period but the overall patterns were confirmed. We believe that the results are not different enough to be added to the manuscript as we now have a extensive number of models included already. However, we attach the results for reference. If the editor or the reviewer disagrees with our decision, we are happy to reconsider.

3. Competing risks: The paper lacks a description of how deaths from other causes are handled. In sensitivity analyses, the authors could also consider assessing potential biasing effects of competing deaths.

In the methods section we added how other causes are treated in our analyses, which now considers all deaths that have occurred during our observation period. Unfortunately we have no information on specific other causes of death for 2020, we only have data on whether a death was associated with COVID-19 or not. In cases where COVID-19 was not identified as the underlying cause of death other causes of death was given as the underlying cause of death. However, we feel that it is out of the scope of this paper to address the very complex issue of competing deaths further, which probably should be done when complete data is available from the National Board of Health and Welfare. However, following your suggestion we extended our analysis with models for all other causes of deaths and thus address some of the associated issues with competing risks directly.

4. Associations with deaths from other causes: Multivariable analyses of associations with deaths from other causes during the same follow up period, similar to the ones presented in Figure 2 and Supplementary Table 2, would strengthen the discussion of how specific the observed associations are with deaths in Covid-19.

Before our initial submission we removed the analyses for all other causes of death from the paper as we felt they were out of the scope of this paper. Following your and the editors suggestions we have reintroduced them. To do this, we had to change the start of the observation period to the day of the first death from COVID-19 (March 13). So, individuals are assumed to be exposed to the risk of dying from COVID-19 and all other causes of death throughout the observation period. The methods, results, and discussion sections have been substantively extended accordingly.

Minor comments

1. Line 53: “The data for this study are 3 135 deaths...”

I would rather say (line 66-67) that the “The data for this study are 6.96 million individuals

(minus those with missing information on key variables) aged 18 and above, of whom 3 122 died from Covid-19 during follow up.”

Following the comments of both reviewers we restructured and rewrote the whole section of Material and Methods from the ground up. After starting with the data sources we now describe the study population and criteria first, followed by the outcome. This should be in line with this comment suggesting to start the data description from the study population and not with the outcome death.

2. Finally, I would check to what extent the associations are the same in “Other counties” (as group) as in Stockholm.

We conducted separate models for Stockholm and all other regions in Sweden. We found very similar mortality differentials across the two models suggesting that the overall level in Stockholm is higher but that the group differences are similar. Excess mortality for non-Swedes as compared to Swedes tend to be slightly more pronounced outside of Stockholm. We still felt that the inclusion would be outside of the scope of the paper and decided to not add the results. Again, we attach the model output for reference.

Reviewer #2

1. It would be helpful to include some quantitative results (HRs and confidence intervals) in the abstract for the main findings reported on lines 10-11.

We understand that some quantitative results might be helpful, however, in this case we decided to follow the journal policies. The author's instructions regarding the abstract state that the abstract should “serve as a brief, non-technical summary of the main results and their implications”. From our reading, presenting hazard ratios and confidence intervals is discouraged, however, we are of course happy to add those point estimates to the abstract if the editor agrees.

2. More information should be provided at the start of the methods on the data sources including links. For example, that the data was accessed via Mona with a link or reference.

This is provided in the reporting summary document but is not explicit in the manuscript. We added more specific information on the data sources at the beginning of the methods section and also added a reference with more detail on the Swedish register system as well as MONA.

3. Description of the eligible population and inclusion criteria (adults aged 18 years and resident in Sweden for 2 years) should be come sooner in the methods section with justification for the inclusion and exclusion decisions taken.

The section on Materials and Methods has been rewritten based on the suggestions of both reviewers. After starting with the data sources we now describe the study population and inclusion and exclusion criteria (also see Figure S1), followed by the outcome.

4. All variables included in the analysis need to be more clearly defined and the authors should give the sources and details of measurement of each of them. Which Swedish administrative registers were used for the data linkage and how were the sociodemographic factors measured and recorded in those registers?

As suggested, the Materials and Methods section now continues with a description of which registers were used and which variables we constructed.

5. The statistical analysis should also present descriptive results including the characteristics of the participants and information on the exposure variables and a summary of follow-up

time.

Following the reviewers' suggestion we have extended the table on the basic characteristics of the participants and exposure variables and moved the table from the appendix into the main text (Table 1), where we now briefly present those results also in the text. As additionally requested by reviewer 1, we also added two tables with even more detailed descriptive results by age groups and sex. They can be found in the supplementary material (Table S1 & S2). As requested by reviewer 2 all supplementary figures and tables are now referred to in the text.

6. The statistical method needs to include statements on a priori decisions taken for the Cox proportional hazard models, such as the reasons why the models were stratified on sex and age. What was the evidence that sex was an effect modifier? Overall analyses should be presented, not just for the sub-groups.

We added the results from the overall analysis to the supplementary material and referred to the table in the results section. We also added further justification for the stratification in the methods section.

7. There is no information provided on model fit or diagnostics. Sociodemographic variables are correlated and so multicollinearity needs to be checked.

We now present model fit and diagnostics in the methods section and also inform on the results of these in the results section.

8. Unadjusted as well as adjusted estimates should be reported.

Following the reviewers' suggestion, we now provide unadjusted and fully adjusted models for the whole study population for COVID-19 deaths (Table S5) and for all other causes of deaths (Table S6). We also reference these tables in the results section.

9. The figures should include a column reporting the actual effect estimates and confidence intervals (with p-values for the overall association for each variable). Alternatively, the authors could consider replacing the figures with tables as it is difficult to read directly from the figures.

We agree that determining the exact value from the figures is difficult, however, the advantage of the figures is that they give a powerful visual of the relative differences between categories within covariates and more easily portray general trends. We now reference the corresponding appendix tables in the text, which include all point estimates and their confidence intervals shown in the figures (Table S1 + S2). In addition, many of the estimates are also reported in the text. The editor also asked to add models for all other causes of death, so we have now two panels in each figure.

10. There is no reference made in the main manuscript to the supplementary figure. Some explanation of its relevance should be included – e.g. to support the choice of time period studied.

Yes, all tables and figures in the supplementary material are now referenced in the text and where applicable we inform about the reasoning behind their inclusion.

11. The paragraph starting at line 164 in the discussion considers other social determinants but there are no references to existing literature.

Yes, we added relevant references.

12. Limitations of the study are not addressed in the discussion.

We added a section on limitations to the end of the discussion.

13. I feel more could be discussed in terms of Sweden's experience with Covid-19 and the approach taken by Sweden to manage Covid-19 and how this underpins the findings of this study. Some description of Sweden's approach in more detail rather than just stating the "more open strategy adopted by Sweden from the outset". Readers will be interested in these results because of the Swedish approach. This is an international journal therefore some more context is needed.

We extended the section on the Swedish context in the end of the introduction and also returned to it in the discussion.

REVIEWERS' COMMENTS

Reviewer #1 (Remarks to the Author):

The authors have provided answers to all my comments on the previous version of the manuscript and have made substantial, appropriate changes both in the main text and in the supplement.

I only have a minor comment left:

1. The authors could consider mention in the abstract that the finding of elevated mortality among immigrants from low and middle-income countries deviates from the findings for other causes of death. All other socioeconomic associations with COVID-19 mortality are more in line with what you see for other causes of death.

Reviewer #2 (Remarks to the Author):

The authors have made significant revisions to the manuscript in line with the reviewers' feedback. This is an elegant analysis demonstrating at a population-wide level the social inequalities of Covid-19 outcomes in Sweden. The work will be of interest globally, and I look forward to seeing it in context as data emerges from studies conducted elsewhere.

I have one small additional comment regarding statistical significance and interpretation. It would be helpful if the tables of cox regression models also include p-values for the overall effect of categorical variables that have more than two categories. If only one categorical class is significant it does not imply that the whole variable is meaningless. For example, in Table S3, the relationship with income tertiles for women is interpreted in the text as no effect, whereas the HR is 1.26 for low versus high (95%CI 1.00-1.58).

Claudia Slimings, Australian National University

Dear Reviewers,

Again, we thank both reviewers for their comments and we are happy that both referees found that our changes have improved the manuscript. In the following our brief reply to each remaining comment:

Reviewer #1:

1. The authors could consider mention in the abstract that the finding of elevated mortality among immigrants from low and middle-income countries deviates from the findings for other causes of death. All other socioeconomic associations with COVID-19 mortality are more in line with what you see for other causes of death.

We agree, the abstract did not reflect the revisions in the best possible way. We now added a sentence and rewrote other parts. The relevant sentence now reads: “By means of individual-level survival analysis we demonstrate that being male, having less individual income, lower education, not being married all independently predict a higher risk of death from COVID-19 and from all other causes of death. Being an immigrant from a low- or middle-income country predicts higher risk of death from COVID-19 but not for all other causes of death.”

Reviewer #2:

I have one small additional comment regarding statistical significance and interpretation. It would be helpful if the tables of cox regression models also include p-values for the overall effect of categorical variables that have more than two categories. If only one categorical class is significant it does not imply that the whole variable is meaningless. For example, in Table S3, the relationship with income tertiles for women is interpreted in the text as no effect, whereas the HR is 1.26 for low versus high (95%CI 1.00-1.58).

We understand the intention of the reviewer and we agree in that if only one categorical class is significant it does not imply that the whole variable is meaningless. We rephrased that part as it does not seem to reflect what we wanted to say (which is that there is no gradient, meaning a continuous decrease in risk the higher the income, but only a difference of the lowest income group relative to the other two groups). However, we also think that adding p-values for the overall effect of categorical variables will complicate our analyses while providing not enough substantive additional insights regarding the identification of disadvantaged subgroups. We therefore decided against the inclusion.